# Green Synthesis of Iron Oxide (Hematite) Nanoparticles and Their Influence on *Sorghum bicolor* Growth under Drought Stress

**DOI:** 10.3390/plants12071425

**Published:** 2023-03-23

**Authors:** Nzumbululo Ndou, Tessia Rakgotho, Mulisa Nkuna, Ibrahima Zan Doumbia, Takalani Mulaudzi, Rachel Fanelwa Ajayi

**Affiliations:** 1Life Sciences Building, Department of Biotechnology, University of the Western Cape, Private Bag X17, Bellville 7535, South Africa; 2SensorLab, Department of Chemical Sciences, University of the Western Cape, Private Bag X17, Bellville 7535, South Africa

**Keywords:** drought, green synthesis, priming, hematite, nanoparticles, *Sorghum bicolor*, oxidative damage

## Abstract

Drought is a major abiotic stress that confronts plant growth and productivity, thus compromising food security. Plants use physiological and biochemical mechanisms to cope with drought stress, but at the expense of growth. Green-synthesized nanoparticles (NPs) have gained great attention in agriculture due to their environmental friendliness and affordability while serving as potential biofertilizers. This study investigates the role of hematite (αFe_2_O_3_) NPs, synthesized from *Aspalathus linearis* (rooibos), to improve *Sorghum bicolor* growth under drought stress. About 18 nm, spherical, and highly agglomerated hematite (αFe_2_O_3_) NPs were obtained. Sorghum seeds were primed with 5, 10, and 15 mg/L αFe_2_O_3_ NPs, and, after seven days of germination, the seedlings were transferred into potting soil, cultivated for fourteen days, and were subsequently water deprived (WD) for a further seven days. A reduction in plant height (78%), fresh (FW; 35%) and dry (DW; 36%) weights, and chlorophyll (chl) content ((total chl (81%), chla (135%), and chlb (1827%)) was observed in WD plants, and this correlated with low nutrients (Mg, Si, P, and K) and alteration in the anatomic structure (epidermis and vascular bundle tissues). Oxidative damage was observed as deep blue (O_2_^●−^) and brown (H_2_O_2_) spots on the leaves of WD plants, in addition to a 25% and 40% increase in oxidative stress markers (H_2_O_2_ and MDA) and osmolytes (proline and total soluble sugars), respectively. Seed priming with 10 mg/L αFe_2_O_3_ NPs improved plant height (70%), FW (56%), DW (34%), total Chl (104%), chla (160%) and chlb (1936%), anatomic structure, and nutrient distribution. Priming with 10 mg/L αFe_2_O_3_ NPs also protected sorghum plants from drought-induced oxidative damage by reducing ROS formation and osmolytes accumulation and prevented biomolecule degradation. The study concludes that green synthesized hematite NPs positively influenced sorghum growth and prevented oxidative damage of biomolecules by improving nutrient uptake and osmoregulation under drought stress.

## 1. Introduction

*Sorghum bicolor* (L.) Moench is a grain-producing cereal crop, ranked fifth in the world and second in Africa after maize. Sorghum grain is mainly used as a source of food for humans in Africa and Asia, whereas sorghum stalks and leaves are used for animal feed and energy production in countries such as Australia, Brazil, and the United States of America [1]. Sorghum has the potential to ensure sustainable food production to combat food insecurity and for use in bioenergy production. Although sorghum is a moderately drought-tolerant crop [2,3], prolonged exposure to drought stress has deleterious effects on its growth, development, and productivity [4].

Drought is a major abiotic stress that affects plant growth and development, but its effects depend on the severity and duration of the stress and the plant species. Drought limits the absorption of water and nutrients in plants and affects plants’ metabolism, stomatal conductance, and photosynthetic rates, and these subsequently decrease growth and negatively impact crop yield [5,6,7]. Water deficit causes the accumulation of reactive oxygen species (ROS) such as superoxide radicals (O_2_^●−^), hydrogen peroxide (H_2_O_2_), and hydroxyl radicals (OH), among others. ROS, when overproduced, are toxic to plant cells since they lead to the damaging of lipid membranes and biomolecules [8]. Drought stress also affects the anatomical structure of plants by causing severe shrinkage of the metaxylem, protoxylem, and phloem [9]. However, as a strategy to increase maximum water supply, plants respond by reducing the vessel diameter while increasing vessel number and thickening the vessel walls [10]. Generally, plants overcome these effects by modification of morphological traits, hormonal responses, osmotic adjustments, and induction of the antioxidant machinery, amongst other mechanisms [11,12] Organic solutes including proline, soluble protein, soluble sugar, and other low-molecular-weight metabolites regulate the osmotic potential of intracellular and extracellular ions in response to osmotic stress, and they also act as ROS scavengers [13]. Conventional fertilizers have been used for years to boost plant growth; however, their overuse has negatively impacted the environment. Thus the use of nanomaterials is a better alternative, due to their low concentration requirement and their effectiveness [14].

Nanotechnology employs smaller particles at the nanoscale level, having at least a dimension between one and a hundred nanometers [15]. The use of nanomaterials holds great promise in agriculture, the nanomaterials serving as plant protectors, nutrition boosters, and alleviators of biotic and abiotic stress effects due to their size-dependent qualities, high surface-to-volume ratio, and unique optical properties [16]. Due to their environmental friendliness, several metal oxide nanoparticles (MONPs), including titanium dioxide (TiO_2_) [17], iron oxide (Fe_3_O_4_) [18], and zinc oxide (ZnO) [19], have attracted significant attention in recent years. Metal oxide NPs have been effectively used in agriculture due to their unique physicochemical properties [20] and the ability to improve growth and development of plants under different growth conditions [21,22,23,24,25,26,27]. Recently, MONPs were found to improve tolerance of *Triticum aestivum* [28,29], *Lallemantia iberica* [30], and *Moringa peregrina* [31] to drought stress by activation of antioxidant enzymes such as superoxide dismutase (SOD), catalase (CAT), and peroxidase (POX).

A few studies have reported on the synthesis of iron nanoparticles [32,33] and their role in crop production under drought stresses [24,25,26,34]. The effects of iron in the alleviation of drought stress might be due to its ability to improve the uptake of water and mineral nutrients and photosynthetic activity and reduce oxidative stress [17,18,28,35]. Green synthesis of nanoparticles must be considered as the preferred technique for plant treatment, due to its several advantages over chemical-based methods. In addition, to low toxicity levels, due to their affordability, green-synthesized NPs are easily scaled up for bulk production [36]. Although there are several kinds of green synthesis methods, the use of plant extracts is a common and the most preferred method, since plants are easily accessible and less toxic than microbes [37,38]. These extracts contain secondary metabolites such as polysaccharides, polyphenolic compounds, amino acids, vitamins, and alkaloids, among other compounds that act as reducing, stabilizing, and capping agents [39,40]. In particular, iron oxide NPs have been synthesized from different plant species including *Lagenaria Siceraria* [17], *Buddleja lindleyana* extract [41], and *Caesalpinia coriaria* (Jacq.) [42], amongst others.

Seed priming is one of the most effective methods to alleviate oxidative stress [43]; this is also because lower concentrations of NPs are used as compared to foliar and soil application. For instance, priming with different NPs effectively alleviated drought stress in Flax plants using iron [44], in wheat using zinc [45], and in Canola plants, where calcium oxide NPs were used [46]. This study is the first to evaluate the role of green hematite (αFe_2_O_3_) NPs synthesized from rooibos tea extracts to improve the growth of *Sorghum bicolor* plants under drought stress. The αFe_2_O_3_ NPs were synthesized and characterized using sophisticated techniques and applied at different concentrations of 5, 10, and 15 mg/L to prime sorghum seeds. Several growth parameters including shoot height, biomass (FW and DW), and physiological traits such as the photosynthetic pigments, anatomical structure, and macronutrient content were assayed. The levels of oxidative stress markers and osmoregulation were also assayed to understand how drought affected sorghum growth at a biochemical level.

## 2. Results

### 2.1. Characterization of Iron Oxide Nanoparticles

The αFe_2_O_3_ nanoparticles (NPs) were successfully synthesized from rooibos tea extracts, and this was observed by a color change to black, which was attributed to a combination of FeCl_3_ (yellow) and rooibos extracts (reddish brown). The extract and the αFe_2_O_3_ NPs indicated UV-vis absorption peaks at 283 nm and 294 nm, respectively (Figure 1A). Fourier-Transform Infrared (FTIR) spectra exhibited strong peaks that can be attributed to various functional groups, which correspond to the phytoconstituents found in the rooibos extract that were responsible for capping and stabilizing the formed NPs (Figure 1B). The spectra of the rooibos extract (black line) and αFe_2_O_3_ NPs (red line) revealed peaks at 3417 cm^−1^ and 3404 cm^−1^ that are attributed to the O-H stretch and a hydroxyl group. The CH_2_ stretch is attributed to peaks at 3241 cm^−1^ and 3254 cm^−1^, and C=O is attributed to peaks at 1626 cm^−1^ and 1620 cm^−1^, where N-O is attributed to peaks at 1519 cm^−1^ and 1577 cm^−1^, C-N is attributed to peaks at 1407 cm^−1^ and 1425 cm^−1^, and the C-O bond is attributed to peaks at 1130 cm^−1^ and 111 cm^−1^. The formation of αFe_2_O_3_ NPs was further confirmed by an FTIR absorption peak at 590 cm^−1^ [47].

The crystallinity, phase, and diameter of the αFe_2_O_4_ NPs, were determined based on the XRD patterns (Figure 1C). Results indicated the formation of crystalline-hematite-phase iron oxide (αFe_2_O_4_) NPs. The XRD pattern was compared with the available phase identification of the Joint Committee on Powder Diffraction Standards (JCPDS card no. 00-019-0629). The peak positions at 23.70° (012), 32.70° (104), 35.10° (110), 40.40° (113), 49.05° (024), 53.60° (116), 57.1° (018), 62.0° (214), 63.70° (300), 71.65° (1010), 77.729° (220), and 77.729° (306) are indexed with the hematite phase of iron oxide NPs. Using the Scherrer formula, as shown in Equation (1), the average particle size of the prepared NPs was calculated to be 18 nm. The lattice plane was confirmed by SAED, which corroborated findings obtained from the SAED lattice patterns (Figure 1D), indicating that they matched well with the XRD patterns, thus confirming the synthesis of hematite (αFe_2_O_3_) NPs.

The αFe_2_O_3_ NPs were spherical in shape and were highly agglomerated as shown on HRTEM images (Figure 2A). The HRTEM histogram data generated using Image J software indicated the size of the hematite (αFe_2_O_3_) NPs to be between 3 and 31 nm (Figure 2B). The value of 16 nm recorded using XRD falls well within this range, further affirming the successful synthesis of the NPs. High-Resolution Scanning Electron Microscopy revealed granular looking and highly-agglomerated NPs (Figure 2C). Energy-Dispersive X-ray Spectroscopy (EDX) was used to calculate the weight composition percentage of the elements found in the NPs. The spectrum indicated a high percentage of iron (Fe) and oxygen (O), confirming the existence of an elemental iron and oxygen signal from the αFe_2_O_3_ NPs with weight compositions of 79.94% (0.21% Sigma) and 18.6% (0.2% Sigma), respectively (Figure 2C,D).

### 2.2. The Effect of αFe_2_O_3_ NP Priming on the Growth Attributes of S. bicolor under Drought Stress

#### 2.2.1. Biomass

The detrimental effects of drought stress in sorghum plants were observed at the morphological level as evidenced by a significant reduction in plant height (78%), fresh weight (35%), and dry weight (36%) in water-deprived (WD) plants in comparison to the control, referred as the well-watered (WW) plants (Figure 3 and Table 1). Interestingly, seed priming with αFe_2_O_3_ NPs protected sorghum plants against severe damage by drought, as seen by a 30%, 70%, and 13% increase in plant height for WD plants grown from seeds primed with 5 mg/L, 10 mg/L, and 15 mg/L αFe_2_O_3_ NPs, respectively (Figure 3). Fresh weight (FW) also increased by 27%, 55%, and 20%, whereas dry weight (DW) increased by 25%, 34%, and 12% in WD plants that were grown from seeds primed with 5 mg/L, 10 mg/L, and 15 mg/L αFe_2_O_3_ NPs, respectively (Table 1).

#### 2.2.2. Photosynthetic Pigment and Growth Parameters

Drought stress severely affected the photosynthetic pigments in the sorghum plants (Table 1). A significant decrease in total chlorophyll (chl) (81%), chla (135%), and chlb (1827%) was observed in the water-deprived (WD) plants. However, priming with αFe_2_O_3_ NPs improved chlorophyll content, with total chl increasing by 68% (5 mg/L αFe_2_O_3_ NPs), 104% (10 mg/L αFe_2_O_3_ NPs), and 55% (15 mg/L αFe_2_O_3_ NPs). Priming also increased chla by 102% (5 mg/L αFe_2_O_3_ NPs), 160% (10 mg/L αFe_2_O_3_ NPs), and 101% (15 mg/L αFe_2_O_3_ NPs), whereas chlb increased by 1162% (5 mg/L αFe_2_O_3_ NPs), 1936% (10 mg/L αFe_2_O_3_ NPs), and 1553% (15 mg/L αFe_2_O_3_ NPs) in WD plants as compared to the unprimed plants (Table 1).

#### 2.2.3. Effect of αFe_2_O_3_ NP Priming on the Anatomic Structure of Sorghum Plants

The anatomical structure, the epidermis and vascular bundle (xylem and phloem) layers, of sorghum plants primed with or without αFe_2_O_3_ NPs was analyzed using HRSEM (Figure 4). Under normal conditions (well-watered; WW), sorghum plants displayed a smooth and clear epidermis as shown in HRSEM micrographs (Figure 4A), whereas drought stress (water-deprivation; WD) severely damaged the anatomical structures as confirmed by the rough and shrunk epidermis (Figure 4B). Iron oxide NPs prevented drought-induced epidermal damage on sorghum shoots resulting in high improvement, showing the epidermis that resembles that of the WW plants (Figure 4C–E). This study further explored the structure of vascular bundle layers to further understand the severity of drought stress in sorghum plants and the protective role of αFe_2_O_3_ NPs (Figure 4F–J). Results revealed large, round, and wider openings of the xylem and phloem (Figure 4F) for WW plants, whereas the xylem of WD sorghum plants revealed oval-shaped walls (Figure 4G). The metaxylem and phloem layers of sorghum plants primed with αFe_2_O_3_ NPs before imposition of drought stress showed improved surface structure and wider and round openings for 5 mg/L and 10 mg/L αFe_2_O_3_ NPs concentrations (Figure 4H–I).

The element distributions of sorghum plants primed with or without αFe_2_O_3_ NPs were analyzed using HRSEM-EDX. Drought and αFe_2_O_3_ NP priming had significant effects on the absorption and transport of macronutrients in sorghum shoots (Figure 5A–E; Table 2). Water deprivation negatively affected uptake and distribution of selected nutrients, and this was reflected by a decrease in the weight percentage of Mg (51.2%), Si (100%), P (88.3%), and K (73%). However, sorghum plants grown from αFe_2_O_3_-NP-primed seeds showed improvement in nutrient uptake and distribution. Magnesium increased by 25.2% (5 mg/L αFe_2_O_3_ NPs), 119% (10 mg/L αFe_2_O_3_ NPs), and 4.8% (15 mg/L α Fe_2_O_3_ NPs), whereas a 100% increase in Si content was observed for all αFe_2_O_3_ NP concentrations. Furthermore, P content increased by 230%, 284%, and 11% for seeds primed with 5 mg/L, 10 mg/L, and 15 mg/L αFe_2_O_3_ NPs, respectively, in WD plants. Similarly, K content increased by 52% (5 mg/L αFe_2_O_3_ NPs), 218% (10 mg/L αFe_2_O_3_ NPs), and 19% (15 mg/L αFe_2_O_3_ NPs).

SEM micrographs for the EDX-investigated area revealed significant morphological changes, seen as severe shrinkages for WD plants (Figure 5G), as compared to a smooth surface area observed in WW plants (Figure 5F). SEM micrographs from WD plants primed with αFe_2_O_3_ NPs revealed smooth epidermis layers with improved surface area and less deformation (Figure 5H–J) for 10 mg/L αFe_2_O_3_ NPs (Figure 5I), showing well-arranged surface structure. 

### 2.3. The Effect of Drought and αFe_2_O_3_ NP Priming on Oxidative Damage in Sorghum

Histochemical staining was performed to determine ROS production as a sign of drought-induced oxidative stress, which led to the damage of lipid membranes and biomolecules (Figure 6A). The NBT staining detected the production of superoxide anion radicals (O_2_^●−^), seen by the clear dark blue spots observed on sorghum leaves (Figure 6A). From the results, it is clear that water deprivation induced O_2_^●−^ production, as more dark blue spots were observed on the leaves of WD sorghum plants as compared to those of well-watered (WW) plants. On the other hand, priming with αFe_2_O_3_ NPs prevented O_2_^●−^ production, as shown by less dark blue spots on the sorghum leaves as compared to unprimed WD plants. In fact, priming with 10 mg/L αFe_2_O_3_ NPs was highly effective, showing less to no spots, indicating the protective benefits of αFe_2_O_3_ NPs against oxidative damage. Compared to WW plants (control), WD (drought-stressed) sorghum plants showed clear injury due to the presence of DAB-stained H_2_O_2_ spots on the surface of the analyzed leaves (Figure 6B). However, priming with αFe_2_O_3_ NPs (5, 10, and 15 mg/L), prevented the over-accumulation of H_2_O_2_ in the leaves, as evident by less intense brown spots, with 5 and 10 mg/L αFe_2_O_3_ NPs showing lesser spots, resembling the leaves of WW plants. However, 15 mg/L αFe_2_O_3_ NPs was less effective in both NBT- and DAB-stained leaves. 

Drought stress significantly increased H_2_O_2_ by 25% and MDA content by 26% as compared to well-watered plants (Figure 6C,D). This increase in the accumulation of oxidative stress markers is an indication of drought-induced oxidative damage of lipid membranes and other biomolecules. Seeds primed with 5, 10, and 15 mg/L αFe_2_O_3_ NPs showed 24, 31, and 25% less H_2_O_2_ content, respectively, as compared to the drought exposed plants. Seed priming with αFe_2_O_3_ NPs also prevented damage on the membrane lipids as seen by decrease in MDA content by 16% (5 mg/L αFe_2_O_3_ NPs), 35% (10 mg/L αFe_2_O_3_ NPs), and 25% (15 mg/L αFe_2_O_3_ NPs).

### 2.4. The Effect of Drought and αFe_2_O_3_ NP Priming on Biomolecules Analyzed by Fourier-Transform InfraRed Spectroscopic (FTIR)

FTIR spectroscopy was used to investigate the effect of drought stress on the damage of biomolecules, including phenolics, carbohydrates, proteins, and lipids (Figure 7). Several peaks were assigned to various functional groups. The peak at 3410 cm^−1^ was attributed to the O-H group, confirming the presence of phenolic compounds. The transmittance percentages showed a vibration at 3226 cm^−1^, attributed to C-H stretching of the alkanes, confirming the presence of aliphatic compounds. The peaks at 2045 cm^−1^ and 1633 cm^−1^ can be attributed to C=C and C=O stretching vibration, indicating the presence of alkanes and alkenes and confirming the presence of carbohydrates. The peak at 1567 cm^−1^ can be attributed to C-F stretching vibration for the alkyl halide group. Proteins were confirmed by the presence of amines, seen by peaks at 1390 cm^−1^ and 1121 cm^−1^ that are attributed to N-O stretching, confirming the presence of pectin and aliphatic amines, whereas peaks at 1082 cm^−1^ and 617 cm^−1^ are attributed to N-H stretching vibration, confirming the presence of proteins due to primary and secondary amines. Shifts and major differences between the FTIR spectrum of WW (black line) plants and WD (red line) plants were observed more visibly at the 3226, 1633, 1390, and 1121 cm^−1^ peaks, suggesting the alterations of the structure of proteins and carbohydrates. These changes in the WD spectrum were restored for plants grown from primed seeds with 5 mg/L (blue) and 10 mg/L (green) αFe_2_O_3_ NPs, as their spectra resembled that of WW plants. However, the spectrum of plants under 15 mg/L αFe_2_O_3_ NPs (magenta) priming resembled that of WD non-primed plants, suggesting that this concentration was a bit high. 

### 2.5. Effect of Drought and αFe_2_O_3_ NP Priming on Osmoregulation in Sorghum

To determine the influence of drought stress and αFe_2_O_3_ NP priming on the osmoregulation of sorghum plants, osmolytes including proline and soluble sugars were analyzed (Figure 8). Proline content was induced by 47% in WD plants as compared to the WW plants (Figure 8A). However, when compared with αFe_2_O_3_-NP-primed plants, proline content declined significantly by 32.45% for 5 mg/L αFe_2_O_3_ NPs, 64.29% for 10 mg/L αFe_2_O_3_ NPs, and 46% for 15 mg/L αFe_2_O_3_-NP-primed WD plants (Figure 8A). Drought stress also induced the accumulation of soluble sugars by 65%, as compared to the WW plants. Priming seeds with αFe_2_O_3_ NPs significantly decreased soluble sugar content by 40% for 5 mg/L αFe_2_O_3_ NPs, 57% for 10 mg/L αFe_2_O_3_ NPs, and 27% for 10 mg/L αFe_2_O_3_-NP-primed WD plants.

### 2.6. Pearson’s Traits Correlations

The relationship between the different traits was analyzed using Pearson’s correlation (Figure 9). The closer the opening of the circle and the deeper the blue color, the more strongly positive the correlation is. On the other hand, the closer the opening of the circle and the deeper the brown color, the stronger the negative correlation among parameters. In this study, fresh weight (FW) was strongly positively correlated with dry weight (DW) (*r* = 0.85), plant height (PH) (*r* = 0.80), chlt (*r* = 0.98), chla (*r* = 0.90), and chlb (*r* = 0.80). The same pattern was observed between DW and PH (*r* = 0.98), chlt (*r* = 0.95), chla (*r* = 0.90), and chlb (*r* = 0.85). A strong positive relationship was recorded between PH and chlt (*r* = 0.75), chla (*r* = 0.75), and chlb (*r* = 0.70). Elements were strongly correlated to growth parameters; for instance, Mg, P, and k were strongly correlated to FW, DW, and PH with *r* > 0.75 and highly correlated to Chlt, Chla, and Chlb with *r* > 0.5. Si was moderately correlated to FW, DW, PH, Chlt, Chla, and Chlb. Elements were highly negatively correlated to oxidative markers (H_2_O_2_ and MDA) and osmolytes (total soluble sugars and proline). Growth parameters (FW, DW, and PH) were strongly negatively correlated with MDA, ROS, proline, and soluble sugar. MDA was strongly positively associated with H_2_O_2_ (ROS) and proline (*r* = 0.8), whereas proline was strongly positively associated with MDA (*r* = 1) and ROS (*r* = 0.98). ROS was only strongly positively correlated with proline (*r* = 0.98) and positively with soluble sugar (*r* = 0.75). 

## 3. Discussion

In the present study, αFe_2_O_3_ NPs were synthesized from *Aspalathus linearis* (rooibos) extract as a capping agent, and ferric chloride was used as a precursor [48,49,50]. The green-synthesized NPs were used to prime sorghum seeds to improve sorghum’s growth under drought stress. Upon mixing the aqueous ferric chloride and the rooibos extracts, a combination of yellow and reddish brown solution immediately turned black as a result of the excitement of plasmic resonance, as described previously [49]. The formation of αFe_2_O_3_ NPs was confirmed by a shift in the absorption peak at ~283 nm (corresponding to the rooibos extract) to ~294 nm, which indicated the formation of αFe_2_O_3_ NPs (Figure 1a), as recorded by [51]. The obtained peaks are in line with the standard absorption peak range of 275–350 nm for iron NPs under UV [48,52].

The physiochemical properties originating from the rooibos extracts that were involved in the synthesis of αFe_2_O_3_ NPs were investigated using FTIR spectroscopy [48]. Based on the FTIR spectrum (Figure 1b), rooibos aqueous extracts contained a complex of natural organic compounds, including phenolics, proteins, and carbohydrates, linked to capping and reducing agents during αFe_2_O_3_ NPs synthesis [53]. The –OH groups are the well-reported chemical bonds found in NPs, as they are responsible for oxidation, while carbonyl and carboxylate groups are involved in nanoparticle stabilization [54,55]. This was further confirmed by the appearance of a new peak at 568 cm^−1^, which is within the expected absorption peak range (400 to 680 cm^−1^) for metal NPs. This absorption peak was attributed to the presence of alpha hematite (αFe_2_O_3_) NPs, as previously reported [51,56].

Transmission electron micrographs revealed spherical, granular, and agglomerated αFe_2_O_3_ NPs, with a size of 18 nm (Figure 2). Analysis with XRD and SAED revealed the NPs to be crystalline in nature, which is common for iron NPs, whereas the JCPDS files also confirmed the synthesized NPs to be hematite NPs. The lattice fringe pattern of the NPs detected by the XRD matched perfectly with those of the SAED obtained from HRTEM analysis. Similarly, previous reports on the synthesis of NPs conducted using different plants, including, eucalyptus leaf extracts [57] and *Moringa oleifera* extracts [58], are in agreement with the results of this study. Thus, these results provide great compulsive evidence of the successful synthesis of hematite (αFe_2_O_3_) NPs. 

Drought is by far the most common and dominant abiotic factor threatening food security, predominately in arid and semi-arid atmospheres [59], where the most drought-sensitive crops such as *Zea mays* are cultivated for consumption [60,61]. This is the first study to report the positive effects of green-synthesized hematite (αFe_2_O_3_) NPs on the growth of sorghum *bicolor* under drought stress. In agriculture, NPs are commonly exogenously applied during irrigation, or mixed with soil [24,26], whereas recent studies documented the foliar application of NPs in agriculture [25,62,63]. However, these methods often require high concentrations of NPs, as high as 200 mg/g of soil [64]. Although other studies reported a positive effect on crop growth even at higher concentrations [65], this defeats the whole advantage of low soil toxicity. Thus, to benefit agriculture while minimizing environmental toxicity, this study demonstrated that seed priming with a low αFe_2_O_3_ NPs (5 mg/L to 15 mg/L NPs) concentration has the ability to improve sorghum growth under drought stress. 

Drought stress significantly reduced the growth of sorghum plants, and this was reflected in reduced plant height, FW, and DW (Figure 3). This reduction in growth is attributed to osmotic stress caused by water deprivation, which affects the absorption and transport of nutrients and water, resulting in a decline in turgidity and cell expansion, hence reducing growth [66,67,68]. However, priming sorghum plants with αFe_2_O_3_ NPs (5, 10, and 15 mg/L) prevented drought-induced growth reduction, with 10 mg/L αFe_2_O_3_ NPs demonstrating the highest growth improvement in sorghum plants. This explains how αFe_2_O_3_ NPs improved water absorption, which led to efficient nutrition uptake. Similar results were observed in barley [69], cotton [70], and *Mentha piperita* L. [14] treated with Fe_2_O_3_. 

Chlorophyll content was assayed to further understand how priming with αFe_2_O_3_ NPs alleviates drought-induced growth inhibition (Table 1) in sorghum plants. Chlorophyll is an important photosynthetic pigment that plays a vital role during photosynthesis and could be a good indication of changes in a plant’s metabolism and health under different environmental conditions [71]. Thus, any interference in photosynthesis might be linked to a disturbance in stomatal conductance and photosynthesis rate and efficiency and a reduction in leaf expansion and hence plant growth [72,73,74]. As seen in this study, drought significantly decreased the chlorophyll content of sorghum plants (Table 1), suggesting that the sorghum’s metabolism was affected. However, priming with αFe_2_O_3_ NPs considerably increased the chlorophyll content of water-deprived (WD) plants, with the highest increase observed for 10 mg/L αFe_2_O_3_ NPs. This observation can be attributed to improved water and nutrient absorption that resulted in adequate physiological accomplishments by sorghum [75]; thus, αFe_2_O_3_ NPs contributed in the restoration of sorghum’s metabolism and health.

The effect of αFe_2_O_3_ NPs on the growth of *Sorghum bicolor* under drought stress was also evaluated based on the anatomic structure (epidermis and vascular bundle (xylem, and phloem) layers). Drought stress caused shrinkage of the epidermis, xylem, and phloem layers (Figure 4). This might be due to limited water and nutrient supply leading to a reduction in growth [76]. Similar observations have been reported where water limitation decreased the diameter of the metaxylem and increased their number as a way of adapting to the stress [10]. The detrimental effects on anatomic structure correlated with the decrease in elements content, such as that of Mg, Si, P, and K (Table 2). This is because these structures play an important role in preventing water loss and facilitate nutrient absorption [77,78]. The considerable decrease in the size of the xylem and phloem layers that resulted from water deprivation might be another reason for decreased photosynthetic pigments [79]. Nutrients play a critical function which is related to enzyme activation, osmotic adjustment, turgor generation, cell expansion, regulation of membrane electric potential, and homeostasis [80]. Ions have been associated with activating pathways that result in better dry weight (DW) of plants, which explains the relation between higher percentage increases in DW and ion contents in WD plants grown from primed seeds [81]. Improved water and nutrients transport shows a strong correlation with improved xylem and phloem structure [82,83,84] and improved plant height as also observed in this study. This is true since similar results were observed in this study, where water deprivation led to a low concentration of elements and deformed/shrunk layers of the xylem and phloem structures as compared to the WW plants.

On the contrary, priming seeds with Fe_2_O_3_ NPs prior to drought stress prevented shrinkage of the epidermis, xylem, and phloem, especially for low concentrations (5 mg/L and 10 mg/L αFe_2_O_3_ NPs). A higher percentage of Mg, Si, P, and K was also observed, and this was attributed to αFe_2_O_3_ NPs facilitating nutrient distribution under drought stress. Furthermore, αFe_2_O_3_ NPs improved the epidermis surface layer, facilitating the opening of the metaxylem and phloem. Hence, it is suggested in this study that αFe_2_O_3_ NPs mitigated the effects of drought by improving micronutrient uptake [85,86].

To understand the biochemical effects of αFe_2_O_3_ NPs in improving the sorghum growth under drought stress, ROS production, integrity of biomolecules, and osmolyte content were investigated. In assessing the level of oxidative damage caused by drought and the ability of αFe_2_O_3_ NPs to prevent oxidative stress, ROS accumulation and MDA content were quantified (Figure 5). Drought caused the over-accumulation of ROS (H_2_O_2_ and O_2_^●−^), and generation of MDA, which led to oxidative stress and hence impaired growth of the plants [87,88,89]. This suggests that priming with αFe_2_O_3_ NPs improved the antioxidant capacity of sorghum plants and is responsible for lower oxidative damage as reported previously in *Zea mays* [90]. This study found a positive correlation between high ROS and proline content accumulation, which explains the lower oxidative damage on sorghum plants primed with αFe_2_O_3_ NPs prior to water deprivation. The decrease in oxidative damage, as seen by reduced ROS, might due to the fact that hematite has the ability to regulate the expression of genes responsible for drought tolerance [91], which ultimately induces the expression of antioxidant enzymes for scavenging ROS [92].

The oxidative damage due to drought stress was further examined by determining any alterations on the integrity or structure of biomolecules such as DNA, proteins, and carbohydrates [93,94] using FTIR spectroscopy (Figure 8). The FTIR spectra revealed significant changes in biomolecules as an indication of degradation due to oxidative stress. This was observed by shifts in several spectral peaks for functional groups that correspond to different biomolecules. Shifts were observed for peaks at 3226, 2045, and 1633 cm^−1^, corresponding to carbohydrates, while peaks at 1390 cm^−1^ and 1121 cm^−1^ corresponded to proteins. Several reports have indicated the deleterious effects of drought-induced oxidative damage on biomolecules [14,95,96]. Sorghum plants grown from Fe_2_O_3_-NP-primed seeds showed an improved FTIR spectra, which resembled that captured from WW (control) samples. The major shifts observed in spectral peaks were reversed when seeds were primed with 5 mg/L (green) and 10 mg/L (blue) αFe_2_O_3_ NPs prior to water deprivation. [91] This indicates that priming with αFe_2_O_3_ NPs reduced oxidative damage and prevented the degradation of biomolecules [97]. 

The osmolytes (proline and soluble sugars) are efficient osmoprotectants that mitigate stress, and their responses under drought stress have been well documented [35,98,99]. In this study, a significant increase in proline and soluble sugar was observed under drought stress (Figure 8) as proof that the plants adopted higher osmotic adjustment to overcome dehydration caused by a lack of water supply [100]. These findings also suggest that proline may have played a role in scavenging H_2_O_2_ and O_2_^●−^ and subsequently boosted the plant’s ability to deal with oxidative damage [101]. On the contrary, priming with αFe_2_O_3_ NPs decreased both proline and soluble sugars in WD plants as compared to the controls (WD plants without priming). Similar observations were reported in iron-coated-with-chitosan NPs in peppermint [102], foliar application in cucumber [103], and in wheat [104]. Soluble sugar has a well-known essential role in metabolism and osmotic adjustment processes and also functions as a substrate in biosynthesis processes, energy production, and also in sensing and signaling systems. A study by [35] showed a correlation between the protection of plants against dehydration and an increase in soluble sugars in shoots and roots. This study suggested that αFe_2_O_3_ NPs facilitated the activation of proline and soluble sugars, but these may have also played a role in stabilizing cellular membranes, maintaining turgor, and working as osmoprotectants. This was further confirmed by the impressive maintenance of the membrane structure of plants as viewed under SEM micrographs (Figure 5). Pearson correlation analysis was performed to assess the potential associations between the different studied experimental traits. A strong positive correlation was observed (Figure 9) among the growth parameters FW and DW and the chlorophyll contents, and a similar trend was also observed in oxidative markers and osmolytes. 

This study demonstrated the positive effects of αFe_2_O_3_ NPs on sorghum growth by alleviation of oxidative damage. There was a strong positive correlation between all studied traits, including elements, growth attributes (PH, DW, and FW), oxidative markers (ROS and MDA), osmolytes (proline and soluble sugars), and chlorophyll contents. A negative correlation was observed between growth attributes (PH, DW, and FW) and biochemical traits (ROS, MDA, osmolytes). The positive correlations between oxidative stress markers and osmolytes may be explained by the reduction of oxidative damage due to the role played by αFe_2_O_3_ NPs in facilitating the prevention of ROS and MDA formation under drought stress, as this could be due to enhancement of efficient water and nutrients use subsequently leading to alleviation of drought stress [75,105]. Most traits were highly strongly correlated, which gives confidence in bringing study traits to close together.

## 4. Materials and Methods

### 4.1. Green Synthesis of Iron Oxide Nanoparticles

#### 4.1.1. Extracts Preparations 

Green extracts were prepared from *Aspalathus linearis* (Rooibos Lager) leaves purchased at Spar supermarket (on 12 August 2019, Bellville, South Africa). About 5 g of tea material was boiled in 250 mL of autoclaved distilled water (ddH_2_O) at 80 °C for two hours. The extracts were then filtered using Whatman filter paper No. 1 and used immediately or kept at 4 °C until further processing [106]. 

#### 4.1.2. Synthesis of αFe_2_O_3_ Nanoparticles

Synthesis of αFe_2_O_3_ nanoparticles (NPs) was done following the previous method [107]. Briefly, 250 mL of 0.1 M anhydrous ferric chloride (FeCl_3_) (Sigma-Aldrich, St. Louis, MO, USA, C451649-1G) was mixed with 250 mL of the green extracts while vigorously stirring at 1000 rpm for 5 h under heat at 80 °C. The reaction mixture was cooled to room temperature, followed by centrifugation at 13,000 rpm for 15 min. The supernatant was discarded while the pellets were resuspended in autoclaved distilled water; this process was repeated three times to remove all the impurities. After washing, the synthesized nanoparticle solution was freeze-dried, and the obtained powder was calcined at 600 °C for 2 h to obtain a more crystallized structure of the NPs. The resulting reddish-brown powder was kept in an airtight container and stored at room temperature until further use.

### 4.2. Characterization of Iron Oxide Nanoparticles

To confirm and characterize the synthesis of the iron oxide nanoparticles (NPs), both the green extracts and the synthesized NPs materials were analyzed. Ultraviolet–visible spectroscopy (Nicolette Evolution 100, Thermo Electron Corporation, Johannesburg, South Africa) was used to verify the formation of the NPs using a wider spectral region/window (200 to 800 nm). The range of the UV bands obtained was between 200–500 nm wavelength. The functional groups of the extract and NPs were examined using Fourier-Transform Infrared (FTIR) spectroscopy (PerkinElmer Spectrum 100-FTIR Spectrophotometer, PerkinElmer (Pty) Ltd., Midrand, South Africa) at a spectral range between 400 and 4000 cm^−1^. The surface structural morphology of the NPs was examined using High-Resolution Scanning Electron Microscopy (Zeiss Auriga HR-SEM; Carl Zeiss Microscopy GmbH, Jena, Germany), and the elemental composition of the NPs was determined on the SEM using Energy-Dispersive X-ray Spectroscopy (EDX) performed on a Zeiss Auriga detector (Oxford Link-ISIS 300, Concord, MA, USA) [108]. High-Resolution Transmission Electron Microscopy (HRTEM) (Tecnai G2 F2O X-Twin HR-TEM; FEI Company, Hillsboro, OR, USA) was used to determine the shape and diameter of the NPs. The phase purity and particle size of the αFe_2_O_3_ NPs were determined using an X-ray diffractometer (Brucker AXS; D8 advanced diffractometer unit Germany. Scherrer’s formula (Formula 1) was used to determine the crystalline size of the synthesized αFe_2_O_3_ NPs.
(1)$$D=\frac{0.9\lambda }{\beta \cos\theta }\;\mathrm{formula}$$where D is the crystallite size, λ is the wavelength of the X-ray used (1.5406 Ǻ), β is the full width at half maximum (FWHM), and θ is the Bragg’s angle.

### 4.3. Sorghum Seeds Preparation and Growth Conditions

Sorghum (*Sorghum bicolor* (L.) Moench) seeds purchased from Agricol, Brackenfell Cape Town, South Africa were disinfected by firstly soaking them in 70% ethanol while shaking on a motor shaker at 600 rpm for one minute, followed by washing thrice with autoclaved distilled water (ddH_2_O). Seeds were then soaked in 5% sodium hypochlorite (bleach) while shaking at 600 rpm for one hour. The bleach was rinsed off by washing the seeds three times with autoclaved distilled H_2_O. Priming of seeds with αFe_2_O_3_ NPs was done by soaking 25 seeds in 250 mL of 0 (mock), 5, 10, and 15 mg/L αFe_2_O_3_ NPs and leaving them shaking at 200 rpm in the dark overnight on a motor shaker. Seeds were then dried under laminar flow and sown on filter paper placed in a plastic container (28 × 21 and 6 cm height) containing 50 mL of autoclaved ddH_2_O. Germination was allowed to take place in the dark at 25°C and monitored for 7 days. Germinated seeds were subsequently sown in wooden boxes (sized: 30 × 30 cm and 7 cm height) containing 2.1 double grow all-purpose organic potting soil and vermiculite (bought from Stodels Garden Center, Eversdal Road, Bellville, Cape Town, SA) and grown in the greenhouse under controlled conditions (26 °C/22 °C day/night; 16 h/8 h light/dark regimes). Watering was done by sprinkling 300 mL of nutrient solution (Dr. Fisher’s Multi feed, 19:8:16 (43), Reg. No./Nr. K5293, Act No./Wet Nr. 36 of/van1947)) every second day for seven days for the plants’ establishment, followed by watering every second day for seven days. Thereafter, plants were divided into three groups consisting of the (1) control (well-watered; WW), (2) drought stressed, plants that are water deprived(to be referred to as water-deprived; WD), (3) plants grown from seeds were primed with NPs (5 mg/L, 10 mg/L and 15 mg/L), and then water deprived. Plants, which served as the control continued to receive water every second day for 7 days. Sorghum plants were harvested on day seven after water deprivation without re-watering; plants were rinsed thoroughly with dH_2_O and stored at −80 °C until further use.

### 4.4. Growth Parameters

Plant height was measured using a ruler in the mm range before harvesting. Shoot fresh weights (FW) were weighed using a Mettle Toledo AE50 analytical balance (Marshall Scientific, Hampton US). To determine dry weights (DW), shoots from which the FW was measured were then oven-dried at 70 °C for 72 h, until a constant weight was attained. To analyze the anatomical structure (epidermis, xylem, and phloem) and element distribution, samples were sent to the Physics Department, University of Western Cape, South Africa and analyzed using High-Resolution Scanning Electron Microscopy (HRSEM) and HRSEM-EDX as previously described [23,109]. All spectra were analyzed using the built-in Oxford Inca software suite. Samples were then imaged and collected using a Tescan MIRA field emission gun scanning electron microscope, operated at an accelerating voltage of 5 kV using an in-lens secondary electron detector.

### 4.5. Determination of Photosynthetic Pigments

Photosynthetic pigments were determined from shoot materials (0.1 g) that were first grinded in liquid nitrogen; the ground materials were incubated in 10 mL of 80% acetone and incubated in the dark for 5 min or until the leaf’s tissues completely lost color. Estimation of chlorophyll (Chl) was performed as reported previously [110]. To determine the Optical Density (OD), the extracted pigments were centrifuged at 15,000 rpm for 15 min at 4 °C, and the absorbance of the supernatant was measured at 645 and 663 nm using a Helios^®^ Epsilon visible 8 nm bandwidth spectrophotometer (Thermo Fisher Scientific, Waltham, MA, USA). The concentration of chlorophyll content was estimated according to the following formulas listed below:
Chla = [12.7 (OD663) − 2.69 (OD645)] × (*V*/1000) × *W*Chlb = [22.9 (OD645) − 4.68 (OD663)] × (*V*/1000) × *W*Chlt = [20.2 (OD645) + 8.02 (OD663)] × (*V*/100) × *W*(2)
where *V* is the volume of sample extract; *W* is the weight of the sample; and Chla, Chlb, and total Chl concentration are expressed as mg g^−1^ FW. 

### 4.6. Analysis of Oxidative Stress Markers

Oxidative stress markers including reactive oxygen species (ROS), such as superoxide (O_2_^●−^), hydrogen peroxide (H_2_O_2_), and malondialdehyde (MDA) content, a marker of lipid peroxidation, were assayed to determine the extent of oxidative damage due to drought stress. All spectrophotometric measurements in this section were done using a Helios^®^ Epsilon visible 8 nm bandwidth spectrophotometer (Thermo Scientific Waltham, MA, USA) unless otherwise stated.

#### 4.6.1. Histochemical Detection of Reactive Oxygen Species

Histochemical detection of ROS was determined as previously described [111]. For detection of superoxide (O_2_^●−^), fresh whole leaves were gently submerged in 0.1% Nitro-blue tetrazolium chloride (NBT) solution and allowed to incubate at room temperature for 12 h in the dark. The formation of H_2_O_2_ was detected by immersing fresh leaves in 1 mg/mL 3′,3′-diaminobenzidine (DAB) solution, and the reaction mixture incubated for 12 h in the dark. After incubation, all histochemical stained samples were boiled in 80% ethanol at 90℃ for 15 min to remove chlorophyll before taking pictures. The presence of dark blue (corresponding to O_2_^●−^) and dark brown (corresponding to H_2_O_2_) stain was captured using a Canon D700 camera (Taiwan/Japan).

#### 4.6.2. Hydrogen Peroxide Content

Quantification of hydrogen peroxide content was performed as described previously by [19,109]. Basically, a 2 mL Eppendorf tube containing 0.25 mL 0.1% trichloroacetic (*w/v*) acid (TCA), 0.50 mL potassium iodide (1 M), and 0.25 mL potassium phosphate buffer (10 mM, Ph 7) was used to homogenize about 100 mg of pulverized sorghum plant material. Centrifuging was done in tubes using a vortex for 15 min at 10,000 rpm (4 °C). Samples were centrifuged and then kept at ambient temperature for 20 min while being shielded from light. A 96-well microtiter plate with samples was used to detect the absorbance at 390 nm. H_2_O_2_ standard solution was used as the quantification method to obtain the calibration curve.

#### 4.6.3. Malondialdehyde Content

The level of lipid peroxidation was determined by measuring malondialdehyde (MDA) content as described previously by [112]. A total of 100 mg of fresh sorghum ground shoot materials was homogenized in 1 mL of 0.1% TCA (*w/v*). Tubes containing the extract were vortexed to mix well and centrifuged at 13,000 rpm (4 °C) for 10 min. Small holes were punched on the cap of new 2 mL Eppendorf tubes using a syringe needle to avoid tubes bursting due to pressure from the heat and reaction by-products. About 400 µL of the supernatant was transferred into a new 2 mL Eppendorf tube containing 1 mL of 0.5% TBA; the reaction was then boiled at 80 °C in a water bath for 30 min. Tubes were chilled on ice for 5 min and centrifuged for another 5 min at 13 500 rpm (4 °C) to precipitate any remaining TBA. A total amount of 200 µL of the supernatant was transferred into 96-well microtiter plates in triplicates to measure the OD at 532 nm and 600 nm. The content of MDA was calculated using formula 3.
MDA (nmol) = D (A 532 nm − A 600 nm)/1.56 × 105(3)

The absorption coefficient for the calculation of MDA is 156 mmol^−1^ cm^−1^.

#### 4.6.4. Fourier-Transform Infrared Spectroscopy (FTIR) Analysis of Biomolecules

The FTIR spectrum of sorghum shoots was analyzed following a method described by Grace et al. (2020) using a PerkinElmer Spectrum 100-FTIR Spectrometer (PerkinElmer (Pty) Ltd., Midrand, South Africa). About 10 mg of dry sorghum shoot tissues were combined with 100 mg of potassium bromide (Kbr), then crushed and ground into a fine powder. The obtained powder was analyzed using a wider window between 400 and 4000 cm^−1^ with 32 scans for a spectrum at 4 cm resolution.

### 4.7. Determination of Osmolyte Content

#### 4.7.1. Proline Content

Proline content was measured as previously described [19]. A total of 100 mg ground shoot material was re-suspended in 500 µL of 3% aqueous sulfosalicylic acid and the reaction mixture was centrifuged at 13,000 rpm for 20 min. About 300 µL of the supernatant was mixed with 600 µL of 2.5% ninhydrin reaction solution and boiled for 10 min at 98 °C. The sample was chilled on ice before it was mixed with equal volumes of pure proline. The optical density was measured at 520 nm using a FLUOstar^®^ Omega (BMG LABTECH, Ortenberg, Germany) microtiter plate reader. The proline content was determined from a standard curve using pure proline as a standard.

#### 4.7.2. Soluble Sugars Content

Soluble sugars were determined as previously described [110], with slight modifications. About 100 mg of ground shoot material was mixed with 10 mL of 80% acetone and incubated in the dark until the leaves completely lost color. One milliliter of supernatant was added to a test tube containing 3 mL of 2% anthrone and the reaction was then boiled at 98 °C for 15 min. The mixture was cooled on ice for 5 min before recording the OD at 625 nm on a Helios^®^ Epsilon visible 8 nm bandwidth spectrophotometer (Thermo Fisher Scientific, Waltham, MA, USA). The total soluble sugar content was estimated by calibrating a standard curve using glucose and the content was expressed as mg g^−1^ FW.

### 4.8. Statistical Analysis

All experiments were repeated at least three times and data were statistically analyzed by a one-way ANOVA using Minitab^®^ 21 Statistical Software (https://www.minitab.com/en-us/support/downloads/, accessed on 7 May 2022). Data in the Figures and Tables represent the mean ± standard deviation (*n* = 3). Statistical significance between control and treated plants was determined by Tukey Test for comparison at a 95% confidence interval. The difference was regarded as significant when *p* < 0.05. Means that do not share a letter in a column were significantly different. Gally and “my norm” from R software was used for Pearson’s correlation (*r*) matrix analysis.

## 5. Conclusions

It is necessary to develop new environmentally and cost-effective methods that have the potential to alleviate the effects of abiotic stress and hence increase food production. Drought stress is responsible for a significant reduction in agricultural productivity, as shown in this study. Drought stress led to a reduction in sorghum growth parameters and photosynthesis and biomolecule damage. In the current study, we successfully synthesized stable αFe_2_O_3_ NPs from rooibos tea extracts. The αFe_2_O_3_ provide a means of efficiently enhancing the physiological and biochemical traits and growth of sorghum by reducing the oxidative stress. Additionally, the administration of αFe_2_O_3_ NPs boosts photosynthesis and water and mineral nutrient use, which strengthens membrane performance such as in xylem and phloem layers. Furthermore, the positive effect of αFe_2_O_3_ NPs was more obvious since it prevented the excessive ROS generation brought on by drought stress. The application of 10 mg/L of αFe_2_O_3_ NPs, as compared to other concentrations examined in this study, proved to be more effective in reducing drought stress. It is therefore recommended to consider the use of green-synthesized hematite NPs in the search for effective bio-stimulants to improve crop yield under environmental stress. However, there is still a need to further investigate αFe_2_O_3_ NPs’ protective effects at the molecular level in cereal crops to better understand αFe_2_O_3_ NPs mechanism of action in ameliorating drought stress. Future work will include investigating the molecular response traits in sorghum that might be induced by αFe_2_O_3_ NPs, and this will assist in deducing the mechanism of action by αFe_2_O_3_ NPs in ameliorating drought stress. 

## Figures and Tables

**Figure 1 plants-12-01425-f001:**
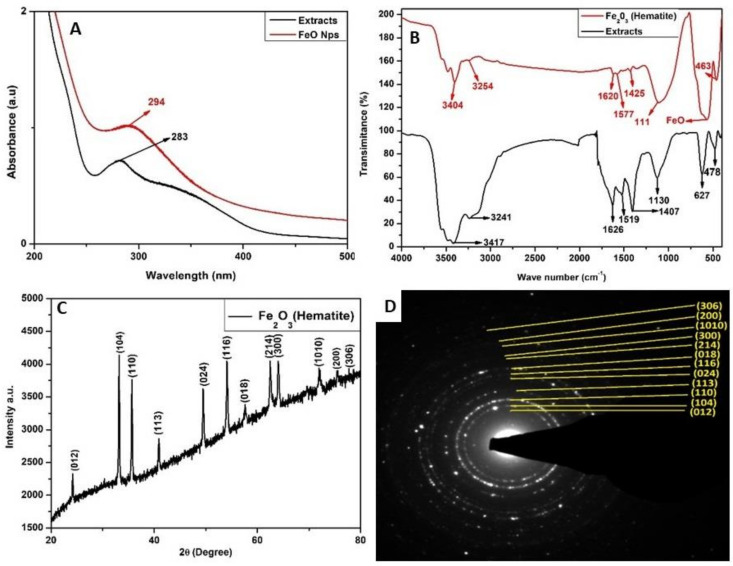
Plasmic resonance and structural characterization of green-synthesized hematite (αFe_2_O_3_) nanoparticles (NPs) using UV-Vs spectroscopy (**A**), FTIR spectra of hematite and rooibos tea extracts (**B**), XRD analysis of the typical Hematite patterns (**C**), and SAED from HRTEM showing lattice planes resembling that of XRD patterns (**D**).

**Figure 2 plants-12-01425-f002:**
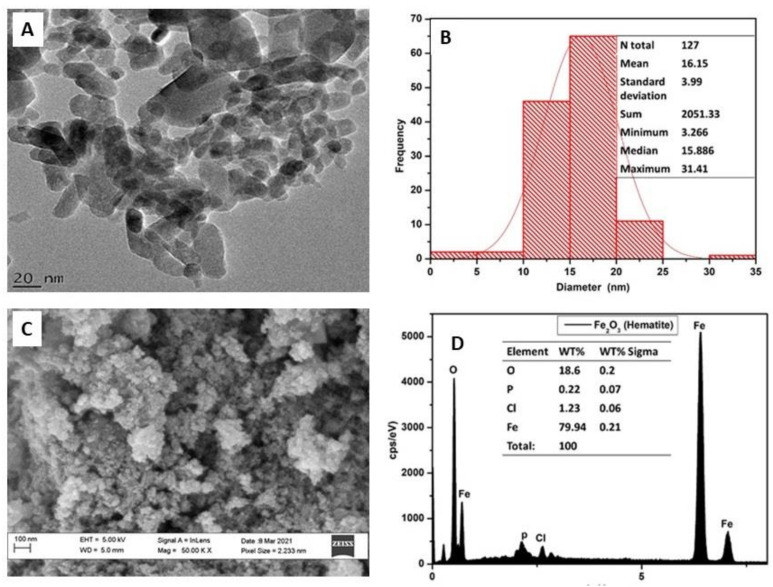
Structure, morphology, and size of green-synthesized αFe_2_O_3_ NPs after annealing at 600 °C for 2 h. HRTEM micrograph showing shape, sizes, and distribution of hematite NPs (**A**), histogram size analysis of NPs (**B**), HRSEM micrograph showing the surface morphology of the synthesized NPs (**C**), and HRSEM-EDX showing weight-percentages composition and distribution of elements in NPs (**D**).

**Figure 3 plants-12-01425-f003:**
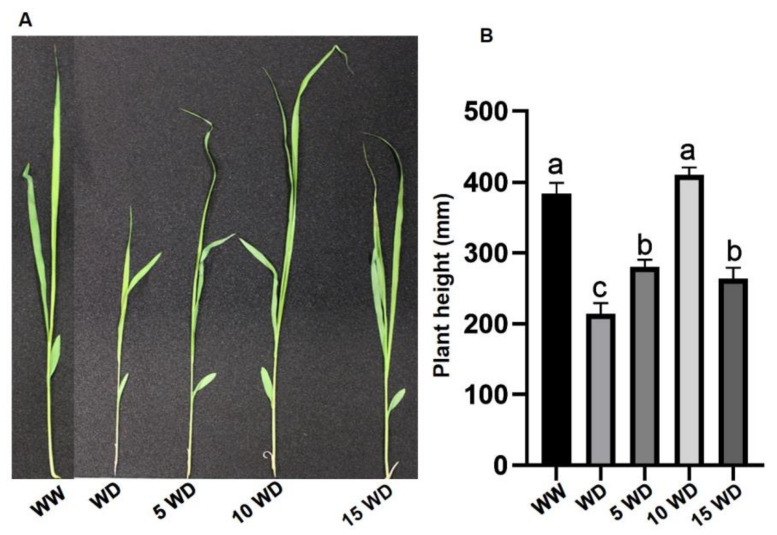
Phenotypic representation (**A**) and plant height (**B**) of sorghum plants under drought stress, grown from seeds primed with αFe_2_O_3_ NPs. WW = well-watered (control); WD = water-deprived (drought stress); 5 WD = water-deprived + 5 mg/L αFe_2_O_3_ NPs; 10 WD = water-deprived + 10 mg/L αFe_2_O_3_ NPs; 15 WD = water-deprived + 15 mg/L αFe_2_O_3_ NPs. Data in the figure represent the mean ± standard deviation (*n* = 3). Different letters indicate significant differences (*p* < 0.05) based on ANOVA one-way variance analysis followed by Tukey’s comparison test.

**Figure 4 plants-12-01425-f004:**
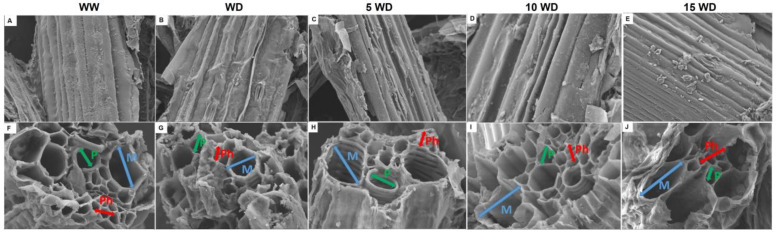
Effect of αFe_2_O_3_ NP priming on the anatomy of sorghum plants under drought stress: epidermis layers ((**A**–**E**); magnification = 10 µm); vascular bundle layers [metaxylem (M; blue), protoxylem (P; green), and phloem (Ph; red)] ((**F**–**J**); magnification = 2 µm). WW =well-watered (control); WD = water-deprived (drought stress); 5 WD = water-deprived + 5 mg/L αFe_2_O_3_ NPs; 10 WD = water-deprived + 10 mg/L αFe_2_O_3_ NPs, 15 WD = water-deprived + 15 mg/L αFe_2_O_3_ NPs Figures were generated on 21 June 2022.2.2.4. Nutrient Uptake.

**Figure 5 plants-12-01425-f005:**
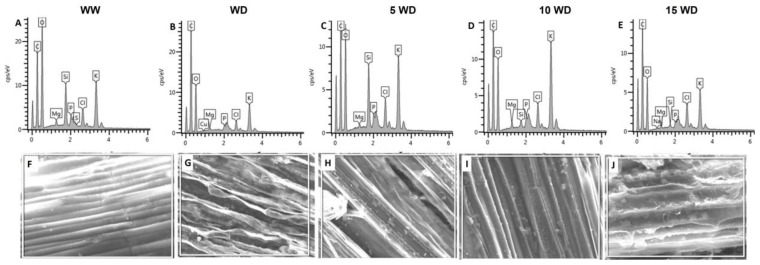
Effect of αFe_2_O_3_ NP priming on the element distribution in sorghum plants under drought stress: EDX spectroscopy showing the element distribution in sorghum plants (**A**–**E**); SEM micrographs for the EDX-investigated area (**F**–**J**). WW = well-watered (control); WD = water-deprived (drought stress); 5 WD = water-deprived + 5 mg/L αFe_2_O_3_ NPs; 10 WD = water-deprived + 10 mg/L αFe_2_O_3_ NPs, 15 WD = water-deprived + 15 mg/L αFe_2_O_3_ NPs. Figures were generated on 21 June 2022.

**Figure 6 plants-12-01425-f006:**
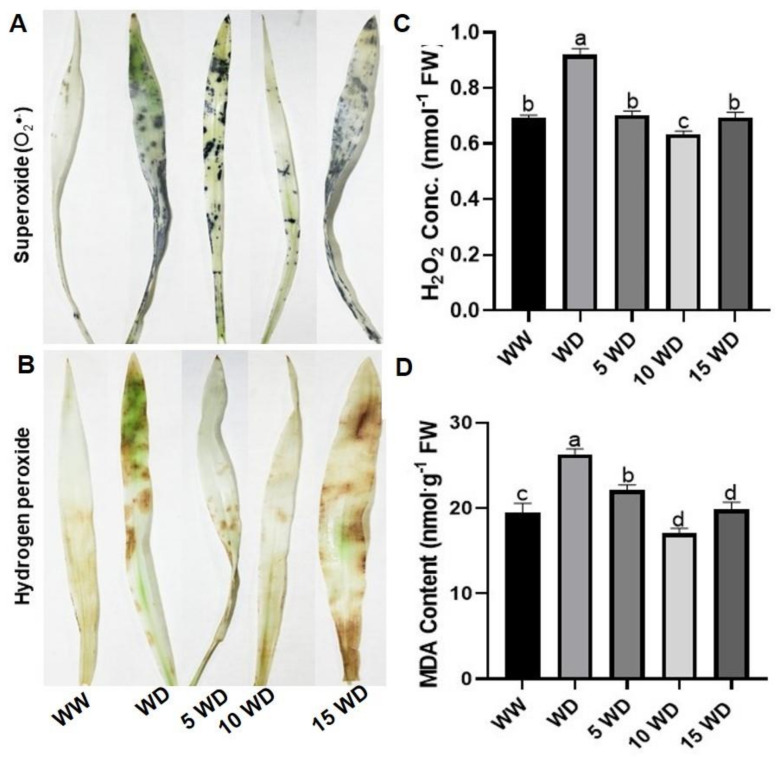
Effect of drought and αFe_2_O_3_ NP priming on oxidative stress markers: Histochemical detection of O_2_^●−^ (**A**) and H_2_O_2_ (**B**); Quantification of H_2_O_2_ (**C**) and MDA content (**D**) in sorghum plants. WW = well-watered (control); WD = water-deprived (drought stress); 5 WD = water-deprived + 5 mg/L αFe_2_O_3_ NPs; 10 WD = water-deprived + 10 mg/L αFe_2_O_3_ NPs, and 15 WD = water-deprived + 15 mg/L αFe_2_O_3_ NPs. Data in the figure represent the mean ± standard deviation (*n* = 3). Different letters indicate significant differences (*p* < 0.05) based on ANOVA one-way variance analysis using Tukey’s comparison test.

**Figure 7 plants-12-01425-f007:**
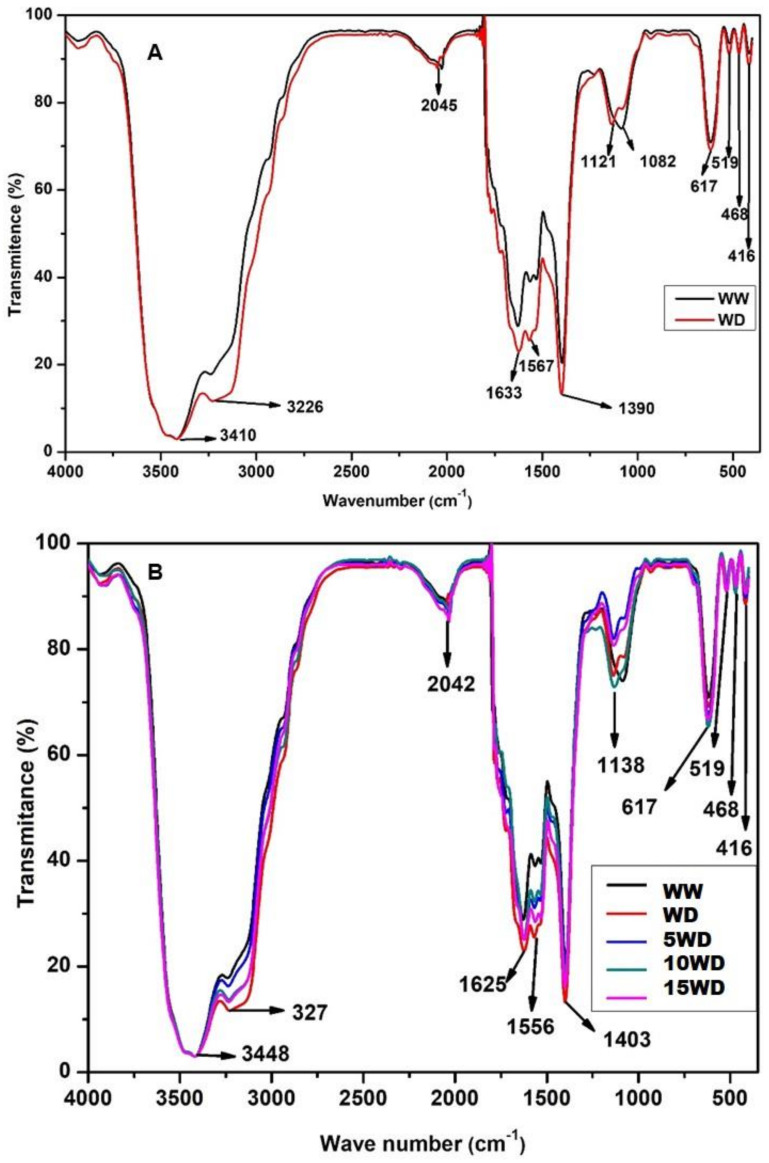
FTIR analysis of the effect of drought (**A**) and αFe_2_O_3_ NP priming (**B**) on biomolecules in sorghum plants. WW (well-watered) = control plants (black), WD (water-deprived) = drought stress (red), 5WD = drought-stressed plants primed with 5 mg/L αFe_2_O_3_ NPs (blue), 10WD = 10 mg/L αFe_2_O_3_ NPs (green) and 15WD = 15 mg/L αFe_2_O_3_ NPs (pink).

**Figure 8 plants-12-01425-f008:**
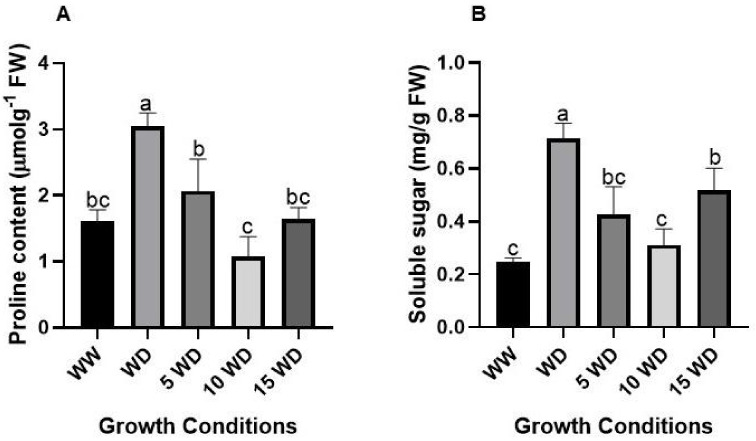
Effect of drought and αFe_2_O_3_ NP priming on osmolytes proline (**A**) and soluble sugar (**B**) content accumulation in sorghum. Error bars represent the SD calculated from three biological replicates. WW = well-watered (control); WD = water-deprived (drought stress); 5 WD = water-deprived + 5 mg/L αFe_2_O_3_ NPs; 10 WD = water-deprived + 10 mg/L αFe2O_3_ NPs, 15 WD = water-deprived + 15 mg/L αFe_2_O_3_ NPs. Data in the figure represent the mean ± standard deviation (*n* = 3). Different letters indicate significant differences (*p* < 0.05) based on ANOVA one-way variance analysis using Tukey’s comparison test.

**Figure 9 plants-12-01425-f009:**
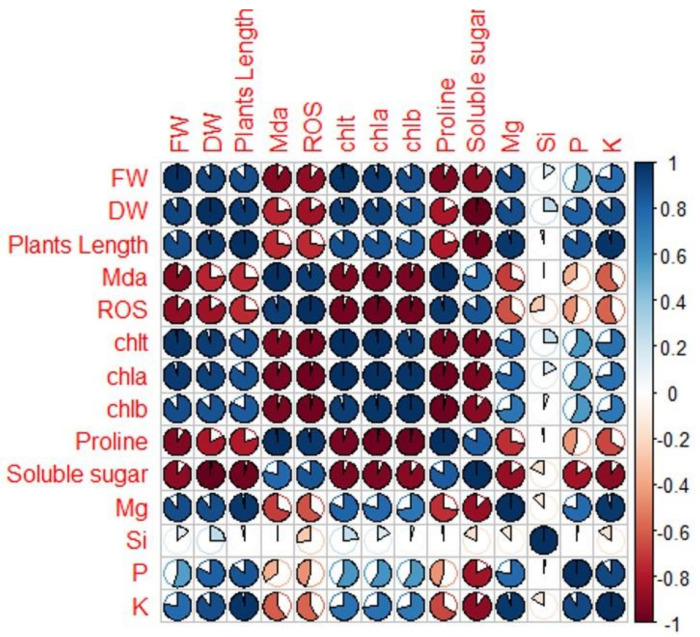
Pearson correlation between different parameters in response to αFe_2_O_3_ NPs treatment under drought stress. Traits included in the analysis are FW (fresh weight), DW (dry weight), PH (plants height), MDA (malondialdehyde), ROS (reactive oxygen species), Chlt (total chlorophyll), Chla (chlorophyll a), Chlb (chlorophyll b), proline, and soluble sugars. The color of the circles and the coefficient values refer to the strength and significance of the correlation: 0.75–1 = strongly correlated; 0.5–0.75 = highly correlated; 0.25–0.50 = moderately correlated; and 0–0.25 = weekly correlated. Under 0 = negatively correlated.

**Table 1 plants-12-01425-t001:** Effect of αFe_2_O_3_ NP priming on the growth parameters of sorghum plants under drought stress. Growth parameters include fresh weight (FW), dry weight (DW), and chlorophyll (Chl) content. Data in the table represent the mean ± standard deviation (*n* = 3). Different letters in the column indicate significant differences (*p* < 0.05) based on ANOVA one-way variance analysis using Tukey’s comparison test.

Growth Conditions	FW (g)	DW (g)	Chla (mg/g FW)	Chlb (mg/g FW)	Chlt (mg/g FW)
WW (control)	0.650 ± 0.046 ^ab^	0.074 ± 0.002 ^a^	9.222 ± 0.892 ^ab^	3.752 ± 0.248 ^a^	11.499 ± 0.55 ^ab^
WD (stressed)	0.481 ± 0.009 ^c^	0.054 ± 0.003 ^c^	3.930 ± 0.101 ^c^	0.195 ± 0.032 ^c^	6.356 ± 0.856 ^c^
5 mg/L αFe_2_O_3_ NPs + WD	0.611 ± 0.024 ^b^	0.068 ± 0.002 ^ab^	7.936 ± 0.958 ^b^	2.458 ± 0.405 ^b^	10.694 ± 0.957 ^b^
10 mg/L αFe_2_O_3_ NPs + WD	0.744 ± 0.045 ^a^	0.073 ± 0.003 ^a^	10.243 ± 0.270 ^a^	3.963 ± 0.388 ^a^	12.982 ± 0.895 ^a^
15 mg/L αFe_2_O_3_ NPs + WD	0.578 ± 0.040 ^bc^	0.061 ± 0.003 ^bc^	7.933 ± 0.066 ^b^	3.218 ± 0.217 ^ab^	9.901 ± 0.121 ^b^

WW = well-watered (control); WD = water-deprived (drought stress).

**Table 2 plants-12-01425-t002:** Overall weight percentage (W%) of element distribution in sorghum plants under drought stress primed with and without αFe_2_O_3_ NPs. WW = well-watered (control); and WD = water-deprived (drought stress); 5 WD = water-deprived + 5 mg/L α Fe_2_O_3_ NPs; 10 WD = water-deprived + 10 mg/L α Fe_2_O_3_ NPs; 15 WD = water-deprived + 15 mg/L α Fe_2_O_3_ NPs. Data in the table represent the mean ± standard deviation (*n* = 3). Different letters indicate significant differences (*p* < 0.05) based on one-way ANOVA using the Tukey’s comparison test.

Growth Condition					
Elements	WW Wt%	WD Wt%	5WD (Wt%)	10WD (Wt%)	15WD (Wt%)
Mg	0.43 ± 0.08 ^a^	0.21 ± 0.008 ^c^	0.263 ± 0.142 ^b^	0.46 ± 0.09 ^a^	0.22 ± 0.05 ^c^
Si	0.28 ± 0.07 ^c^	0 ± 0 ^d^	2.8 ± 0.973 ^a^	0.52 ± 0.02 ^b^	0.22 ± 0.08 ^c^
P	1.37 ± 0.01 ^a^	0.16 ± 0.069 ^c^	0.53 ± 0.055 ^b^	0.62 ± 0.02 ^b^	0.18 ± 0.078 ^c^
K	15.45 ± 0.11 ^a^	4.135 ± 2.240 ^c^	6.295 ± 3.204b ^c^	12.7 ± 0.65 ^ab^	4.9 ± 2.63 ^c^

WW = well-watered (control); WD = water-deprived (drought stress).

## Data Availability

We encourage all authors of articles published in MDPI journals to share their research data. In this section, please provide details regarding where data supporting reported results can be found, including links to publicly archived datasets analyzed or generated during the study. Where no new data were created, or where data is unavailable due to privacy or ethical restrictions, a statement is still required. Suggested Data Availability Statements are available in the section “MDPI Research Data Policies” at https://www.mdpi.com/ethics, accessed on 15 March 2023.

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
