# Peer review of "Green Synthesis of Iron Oxide (Hematite) Nanoparticles and Their Influence on Sorghum bicolor Growth under Drought Stress"

_plants, 2023, doi:10.3390/plants12071425_

Round 1
Reviewer 1 Report
Overall, the article "Green-synthesis of iron oxide (hematite) nanoparticles and 2 their influence on Sorghum bicolor growth under drought stress" is of high quality.
The methods are sound and the authors have done an impressive amout of work.
A bit more explanation in the intrdouction would be required. See also https://www.mdpi.com/2223-7747/10/8/1730
(1) The xylem phenotype is very interesting. How does this compare to the phenotypes observed in Arabidopsis exposed to water scarcity?
See and cite:
Continuous root xylem formation and vascular acclimation to water deficit involves endodermal ABA signalling via miR165
P Ramachandran, G Wang, F Augstein, J de Vries, A Carlsbecker
Development 145 (3), dev159202
(2) Water scarcity is a stressor that all land plants face. Please briefly elaborate how this has shaped their (central and specialized) metabolism across evolution.
See and cite:
Crossroads in the evolution of plant specialized metabolism
Seminars in Cell & Developmental Biology
Volume 134, 30 January 2023, Pages 37-58
The authors show a couple of bar charts. Did they test for normality of their data (Shapiro Wilk test)?
Please plot the individual data points as scatters onto those bars so that we can see the distribution.
Regarding Figure 9: how do the euclidian distances look like?
Figure legend of Figures 1 and 2 are very short. Please explain more here.
Author Response
Dear academic editor, Plants Journal, mdpi
We are grateful for the reviewer’s comments, which have significantly improved our manuscript. We have implemented all the suggested changes and below is a point to point response:
Comments from Reviewer 1, supported by the author’s point-by-point responses
- A bit explanation on the introduction would be required.
Authors Reply: The introduction has been improved and references have been added as suggested by the reviewer:
Line 70-73: Due to their environmental friendliness several metal oxide nanoparticles (MONPs), including titanium dioxide (TiO2) [17], iron oxide (Fe3O4) [18], and zinc oxide (ZnO) [19], have attracted significant attention in recent years.
Line 78-81: Recently MONPs have been found to improve tolerance of Triticum aestivum [28, 29], Lallemantia iberica [30], and Moringa peregrina [31] to drought stress by activation of antioxidant enzymes such as Superoxide dismutase (SOD), catalase (CAT), and peroxidase (POX).
Line 82- 84: A few studies have reported on the synthesis of iron nanoparticles [32, 33] and their role in crop production under drought stresses [24-26, 34].
Line 88-90: In addition, to low toxicity levels, due to their affordability, green-synthesized NPs are easily scaled up for bulk production and affordability [36].
Line 94-95: In particular iron oxide NPs have been synthesized from different plant species including Lagenaria Siceraria [17], Buddleja lindleyana Extract [41], Caesalpinia coriaria (Jacq.) [42] amongst others.
Line 100-103: For instance priming with different NPs effectively to Alleviated drought stress in Flax plants using iron [44], in wheat using zinc [45], and in Canola plants, calcium oxide NPs were used [46].
Citation included
- Kamalizadeh, M., M. Bihamta, and A. Zarei, Drought stress and TiO2 nanoparticles affect the composition of different active compounds in the Moldavian dragonhead plant. Acta physiologiae plantarum, 2019. 41(2): p. 21.
- Rakgotho, T., et al., Green-synthesized zinc oxide nanoparticles mitigate salt stress in Sorghum bicolor. Agriculture, 2022. 12(5): p. 597.
- Rui, M., et al., Iron oxide nanoparticles as a potential iron fertilizer for peanut (Arachis hypogaea). Frontiers in plant science, 2016. 7: p. 815.
- Havas, F. and N. Ghaderi, Application of iron nanoparticles and salicylic acid in in vitro culture of strawberries (Fragaria× ananassa Duch.) to cope with drought stress. Plant Cell, Tissue and Organ Culture (PCTOC), 2018. 132(3): p. 511-523.
- Adrees, M., et al., Foliar exposure of zinc oxide nanoparticles improved the growth of wheat (Triticum aestivum L.) and decreased cadmium concentration in grains under simultaneous Cd and water deficient stress. Ecotoxicology and Environmental Safety, 2021. 208: p. 111627.
- Shoarian, N., et al., Titanium dioxide nanoparticles increase resistance of l. iberica to drought stress due to increased accumulation of protective antioxidants. Iranian Journal of Plant Physiology, 2020. 10(4): p. 3343-3354.
- Foroutan, L., et al., The effects of zinc oxide nanoparticles on enzymatic and osmoprotectant alternations in different Moringa peregrina populations under drought stress. International Journal of Basic Science in Medicine, 2018. 3(4): p. 178-187.
- Water Scarcity is a stressor that all land face. Please briefly elaborate how this has shaped their (central and specialized) metabolism evolution.
Authors Reply: In a quest to reply to this comment and still within the scope of this study, authors have added more information on water scarcity under the introduction and included new citations.
Line 52-57: Drought stress also affects the anatomical structure of plants by causing severe shrinkage of metaxylems, protoxylem and phloem [9]. But as a strategy to increase maximum water supply plants, respond by reducing the vessel diameter, while increasing vessel number and thickening the vessel walls [10].
Citations included.
- Ramachandran, P., et al., Continuous root xylem formation and vascular acclimation to water deficit involves endodermal ABA signalling via miR165. Development, 2018. 145(3): p. dev159202.
- Foyer, C.H. and G. Noctor, Redox homeostasis and antioxidant signaling: a metabolic interface between stress perception and physiological responses. The plant cell, 2005. 17(7): p. 1866-1875.
- The author shows a couple of bar charts. Did they test for normality of their data (Shapiro test)?
Authors Reply: The data was tested and passed for Shapiro wilk normality test using GraphPad prism 9. Figures are included as supplementary data.
- Please plot the individual data points as scatters onto those bars so that we can see the distribution.
Authors Reply: Data plotted and the results have been included as supplementary figures. If the reviewer will like the scatter plots to be implemented in the main manuscript, please let us known
- Regarding the Figure 9: how do the Euclidian distance look like?
Authors Reply: The Euclidian distance looks identical with moderate similarly since the measurements are close to 0.
- Figure legends of Figures 1 and 2 are very short. Please explain more here.
Authors Reply: Explanation added.
Figure 1. Plasmic resonance and structural characterization of green-synthesized hematite (αFe2O3) nanoparticles (NPs) using UV-Vs spectroscopy (A), FTIR spectra of Hematite and Rooibos tea extracts (B), XRD analysis of the typical Hematite patterns (C) SAED from HRTEM showing lattice planes resembling that of XRD patterns (D).
Figure 2. Structure, morphology, and size of green-synthesized αFe2O3 NPs after annealing at 600°C for 2 hours. HRTEM micrograph showing shape, sizes and distribution of Hematite NPs(A), Histogram size analysis of NPs (B), HRSEM micrograph showing the surface morphology of the synthesized NPs (C) and HRSEM-EDX showing weight percentages composition and distribution of elements in NPs (D).

Reviewer 2 Report
The manuscript entitled ’Green-synthesis of iron oxide (hematite) nanoparticles and their influence on Sorghum bicolor growth under drought stress (no. 2249464)’ was written about revealing the potential of using hematite (αFe2O3) nanoparticles (NP) synthesized from rooibos plant to improve sorghum growth under drought stress. Whilst drought significantly reduced sorghum’s growth, nutrients uptake and chlorophyll content due to the increased oxidative damage.
Authors found that seed priming with 10 mg/l αFe2O3 NPs of sorghum improved plant height, fresh and dry weight, total chlorophyll content. The priming also reduced the accumulation of osmolytes and protected biomolecules from degradation by reducing ROS formation.
They measured basic physiologiocal and biochemical variables only to answers their question. The idea, quality and information of microscopy is desirable. The quality and number of figures are ready. Based on the provided data and results, I believe that green synthesized hematite NPs may have a positive influence on sorghum growth and can prevent oxidative damage of biomolecules by improving nutrient uptake and osmoregulation.
Some of my minor comments can be seen below:
- If panels of figures were indicated by capital letters, use capital font on the text of figure legends, too
- The missing of any investigation photosynthetic performance is disadvantageous
- It is not a big mistake, however positive correlation is generally marked by red and negative by blue (or green) colors (Fig. 9)
- Authors should explain or at least discuss using foreign literature, how hematite NPs act and enhancing antioxidant defense systems and regulating the expression of stress-related genes, and how they modulate the expression of genes involved in stress response pathways
Overall, I recommend this study to publish in Plants after some minor revision.
Author Response
Dear academic editor, Plants Journal, mdpi
We are grateful for the reviewer’s comments, which have significantly improved our manuscript. We have implemented all the suggested changes and below is a point to point response:
Comments from Reviewer 2, supported by the author’s point-by-point responses
- If panels of figure were indicated by capital letters, use capital font on the text of figure legends too.
Authors Reply: The font for letters in the figure legends have been changed to match those in the figure panels.
- The missing of any investigation of photosynthetic performance is disadvantageous.
Authors Reply: Thank you for the comment, according to photosynthetic performance, which includes, the photosynthesis rate, stomata conductance, and transpiration, in addition to chlorophyll content is considered a standard technique for studies on drought stress. For the current study, only the chlorophyll content was analyzed to understand the photosynthetic performance in sorghum plants primed with and without αFe2O3 NPs under drought stress. We are happy to say that αFe2O3 NPs improved the photosynthetic performance in sorghum plants under drought stress, as the Chlorophyll content data corroborated with the heights of the plants.
Citation included: Jia, Yuying, Wanxin Xiao, Yusheng Ye, Xiaolin Wang, Xiaoli Liu, Guohong Wang, Gang Li, and Yanbo Wang. "Response of photosynthetic performance to drought duration and re-watering in maize." Agronomy 10, no. 4 (2020): 533. doi:10.3390/agronomy10040533.
- It is not a big mistake however positive correlation is generally marked by red and negative by blue (or green) colours (fig. 9)
Authors Reply: We take the comment in to consideration, how ever we followed the default settings and guidelines of R program, which shows positive correlation in Blue and Negative in red.
- Author should explain or at least discuss using foreign literature, how hematite NPs acts and enhance antioxidant defence system and regulating the expression of stress-related genes, and how they modulate the expression of genes involved in stress response pathways
Authors Reply: We would like to thank the reviewer for this comment, we have added a small description in the discussion section, however being very careful since the current study only focussed on the growth, looking at the morphological and physiological attributes and not much on the biochemical and molecular aspect. However, in the future this will be taken into consideration.
Sentence added in the discussion “ The decrease in oxidative damage as seen by reduced ROS, might due to the fact that Hematite has the ability to regulated the expression of genes responsible for drought tolerance [1], which ultimately induced the expression of antioxidant enzymes for scavenging ROS [2].”
Citations included
- Linh, T.M., et al., Metal-based nanoparticles enhance drought tolerance in soybean. Journal of Nanomaterials, 2020. 2020: p. 1-13.
- Ali, S., A. Mehmood, and N. Khan, Uptake, translocation, and consequences of nanomaterials on plant growth and stress adaptation. Journal of Nanomaterials, 2021. 2021: p. 1-17.